# Landscape Planning for Conservation: The Case of the Flora and Fauna Protection Area "Sierra de San Miguelito", San Luis Potosi, Mexico

Gerardo A. Hernández [1], Fernando A. Rosete [1,*], Lidia Salas [1], Luis F. Alvarado [1], Juan Martinez [2] and José F. Sanchez [1]

1 Escuela Nacional de Estudios Superiores unidad Morelia, Universidad Nacional Autónoma de México, 8701 Antigua Carretera a Pátzcuaro, Exhacienda de San José de la Huerta, Morelia 58190, Mexico; ghercendejas@enesmorelia.unam.mx (G.A.H.); lsalas@enesmorelia.unam.mx (L.S.); fernando.alvarador@enesmorelia.unam.mx (L.F.A.); fsancheze411@gmail.com (J.F.S.)

2 Instituto de Investigaciones en Ecosistemas y Sustentabilidad, Universidad Nacional Autónoma de México, 8701 Antigua Carretera a Pátzcuaro, Exhacienda de San José de la Huerta, Morelia 58190, Mexico; jmc@iies.unam.mx

* Correspondence: fernando.rosetev@enesmorelia.unam.mx

**Abstract:** A supporting study was developed to identify the priority elements for conservation in the region called "Sierra de San Miguelito" (SSM), in the San Luis Potosi State (SLP), Mexico, with the purpose of establishing a federal protected natural area (PNA). The methodological approach used was the integral-spatial analysis applied in territorial planning processes. The study showed that the forests, xerophilous scrubland, and natural grasslands of the SSM present a high biodiversity, an abundance of endemism (27% of species are endemic to the country, *n* = 285), and protected species (5% of reported species). In addition, 32.74% of vertebrates and 18.32% of flora reported for SLP status is present in SSM, with an area that represents only 1.79% of the state territory. As a result of the study, an area of 109,638.95 ha was proposed to be declared a federally PNA. The area provides environmental services that favor the San Luis Potosí city (SLPc) and the surrounding population; therefore, its conservation will promote the preservation of natural, cultural, and landscape heritage, being a transversal axis for sustainable development in its area of influence. The result was the basis for starting the negotiation process, developed in 2021, for the creation of the PNA.

**Keywords:** protected natural area; sustainable development; semi-arid ecosystems; environmental services; biodiversity; cultural heritage



## 1. Introduction

Productive human activities that generate a large-scale ecological crisis are currently identified, manifesting in the occurrence of a series of environmental changes at the global level [1]. These changes occur at a local scale but are so generalized that they have global consequences, such as the loss and degradation of ecosystems, loss of biodiversity, etc. [2]. Mexico has developed a series of public policies on environmental issues to address this crisis and join the fulfillment of the Sustainable Development Goals (SDGs) defined in the United Nations 2030 Agenda [3], within the framework of different strategies and institutional development programs, including the Climate Change Strategy from Protected Natural Areas: 2015–2020 [4]. The conservation and restoration of natural ecosystems are one of the main axes of the environmental policies of this program since these measures are recognized as fundamental for the maintenance of biological diversity, the provision of environmental services for humans, and face climate change through mitigation, adaptation, and vulnerability reduction measures [4].

One of the conservation objectives in Mexico is to preserve at least 10% of the surface of each of the 39 level IV terrestrial ecoregions of the country [5,6]. Currently, the existing

protected natural areas (PNA) cover less than this 10%; so, to achieve the desired goal, it is necessary to increase the PNA through protection schemes. The ecoregions "Piedemontes and plains with grassland, xerophilous scrub and oak, and coniferous forests" and the "Plains of the Zacatecano-Potosino Altiplano with microphilous-crasicaule xerophilous scrub" are considered among the high priority ecoregions for conservation due to their scarce protected area, high biological diversity and high risk of degradation, with some protection regime of 0.8% and 2.8%, respectively [6]. These ecoregions have significant problems of water scarcity [7,8] and desertification [9,10], which may intensify in the future due to climate change and increased soil degradation [8,11–13].

Sierra de San Miguelito (SSM) in San Luis Potosi State (SLP) is a natural area located within the referred to above ecoregions indicated as important for conservation [14]. This mountain range has been recognized for having great ecological importance due to its relatively good conservation status, the high biological diversity it harbors, and for being a source of varying environmental services for the San Luis Potosí city (SLPc) and surrounding population [6,14]. The SSM is characterized by an abrupt and irregular relief, with elevations between 2100 and 2800 m above sea level (masl) and dominated by slopes greater than 30°. The surface hydrology of this area is characterized by an extensive network of intermittent streams that recharge the various dams in the region [15], such as the San José dam and the Cañada de Lobos dam that supply water to the state capital [16,17]. Groundwater hydrology plays a role in the recharge of three aquifers, including that of the SLP city valley [18–21].

This study shows the biophysical and cultural richness basis considered in the elaboration of the technical proposal for the delimitation of the future federal PNA SSM, elaborated from a landscape planning process. In addition to the existing biological diversity, the ecosystem services provided by the area [22–24], the establishment of biological corridors [23,25] and the incorporation of productive activities into the conservation program [26–28] were the main elements to be considered in the landscape planning process that generated the proposal.

The article has the following structure: an introduction with the general problems addressed and the theoretical concepts that support the research, the materials and methods used, the results obtained, a discussion on the relevance of the results and the conclusions of the research.

## 2. Materials and Methods

### 2.1. Study Area

The area proposed as a protected natural one SSM is located in the southwestern region of the San Luis Potosi State and is 109,638.95 hectares, which corresponds to 1.79% of the total area of the state. This portion is distributed among the municipalities of Mexquitic de Carmona, San Luis Potosi, Villa de Arriaga, and Villa de Reyes (see Figure 1). The area presents almost 80% of rugged relief, while the remaining land corresponds to plains and plain areas. These variations associated with the geographic location, climatic conditions, and altitude favor the presence of temperate forest ecosystems and xerophytic scrub characteristics of arid regions.

The following activities are carried out within the SSM: cattle ranching, forestry, firewood collection, hiking, and mining. Once the PNA proposal is approved, it will be important that the management plan includes these activities with a focus on sustainable resource management.

Dry climates are a characteristic of this region [29]. A total of 92.64% of the area is characterized by a semi-arid temperate climate with summer rains and an average annual rainfall of 400–500 mm, while 7.36% corresponds to an arid temperate climate with summer rains and an average annual rainfall of 300–400 mm. The average annual temperature of the region is 12–18 °C [29], and according to historical records of the National Water Commission (CONAGUA) in the northern part of the study area, maximum temperatures of up to 48 °C and minimum temperatures of −10 °C have been recorded [30].

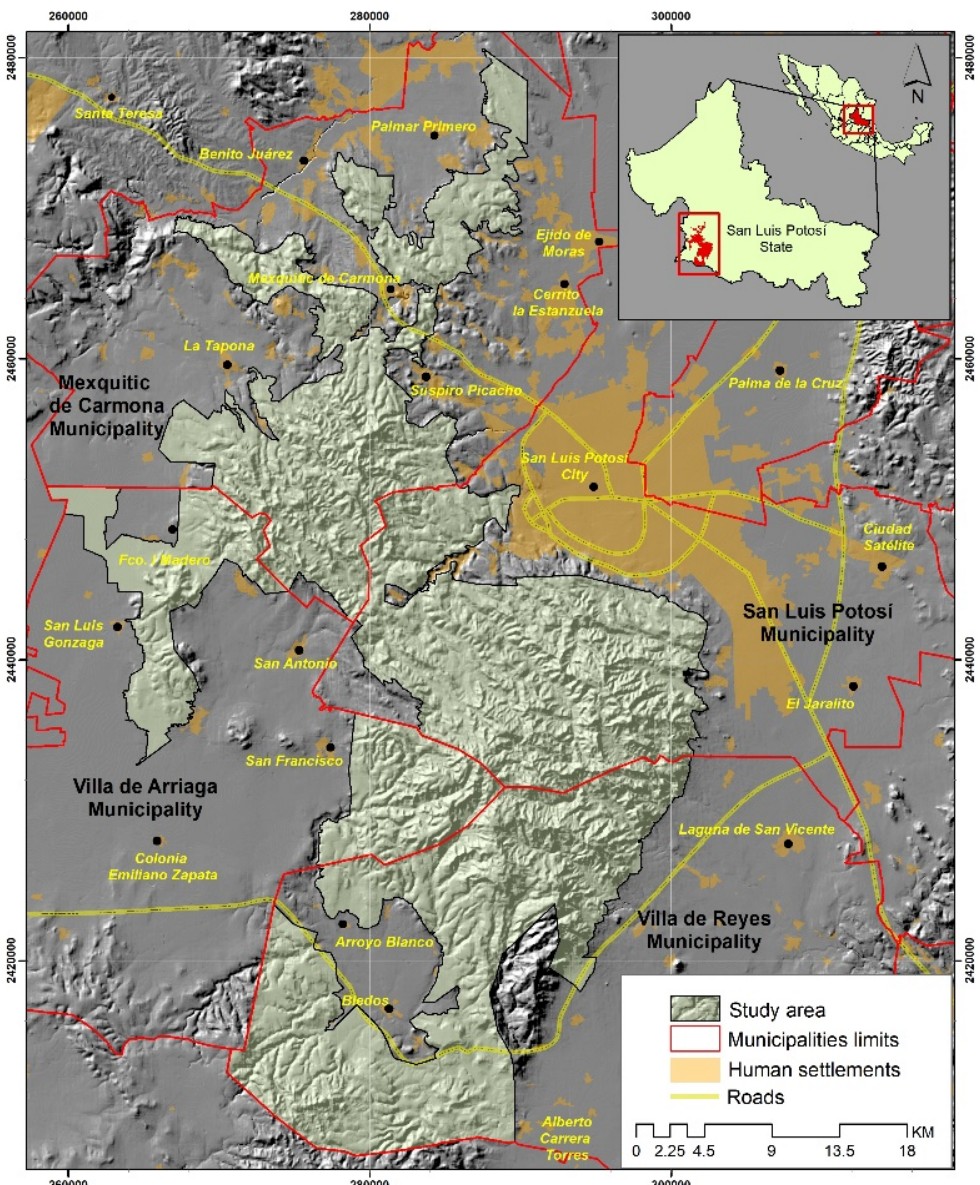

**Figure 1.** Location of the study area.

This climate characteristics favor forested ecosystems with pine species in the higher elevations such as *Pinus cembroides* and *Pinus strobiformisrmis*; oaks such as *Quercus potosina*, *Quercus crassifolia*, *Quercus microphylla*. On the other hand, in the lower areas, xerophytic scrub is identified, where species, such as *Yucca filifera*, *Stenocactus dichroacanthus*, *Stenocactus ochoterenianus*, and different types of agaves and opuntias [31]. It is important to note that in the areas adjacent to the mentioned region, there is a large land with induced pastures and crops that evidence the pressure of human activities in the region.

Forests of the SSM play an important role in providing environmental services by recharging groundwater [32,33], carbon cycle regulation [34–36], contributing to soil formation and retention [37], additionally to reach surrounding populations with timber and non-timber resources for local consumption and commercialization, thus being important as suppliers of environmental provisioning services [14,38,39].

Semiarid ecosystems are highly vulnerable to climate change (precipitation and temperature) [40], as well as highly vulnerable to landscape degradation due to poor agricultural and livestock management [32,41]. Given the biological and ecological importance of

the SSM for SLP and the country, this paper presents the analysis of the information carried out to justify the creation of a PNA under federal jurisdiction in this territory.

*2.2. Data Collection*

The technical study was prepared according to guidelines specified in the terms of reference for the preparation of prior justification studies for the establishment of Protected Natural Areas under the jurisdiction of the Federation, devised by the National Commission of Protected Natural Areas (CONANP) [42]. This document takes up the existing methodological proposals for the elaboration of landscape planning [43–45]. Several factors are considered, including the state of conservation of the ecosystems, the environmental services they provide, their biological richness, the presence of endemic and endangered species, and the strategic natural resources existing at the regional and national levels. Consideration of the social and economic context conducive to the declaration of protected areas is indispensable [42].

2.2.1. Documentary Review

Research in specialized bibliography was carried out to know the characteristics of the study area; scientific journals, bachelor, master, and doctoral theses, technical reports, books, etc., that had previous records of flora, fauna, productive activities, and cultural elements in the study area. Additionally, the database of the National Biodiversity Information System (SNIB) provided by the National Commission for Knowledge and Use of Biodiversity (CONABIO) [31] was consulted.

2.2.2. Land Cover and Land Use Interpretation

The visual classification method was implemented to map the cover land use and vegetation of the SSM PNA. This method consists of digitizing an image taking its characteristics (shape, color, texture) as a reference. For this purpose, a Sentinel image dated 3 March 2020, was used as the basis for adjusting the map corresponding to the VI series of land use and vegetation of the National Institute of Statistics and Geography (INEGI). The classification scale was 1:50,000, which allowed for greater detail and adjustment of the different categories.

2.2.3. Sentinel Image Processing

The image used as a basis for the classification of vegetation cover and land use was downloaded from the Copernicus server (Earth observation program of the European Union), which provided us with the orthorectified and atmospherically corrected scene.

The bands used during the interpretation process were blue (B2), green (B3), red (B4) and near-infrared (B8), all with a resolution of 10 m, then fake color composites were made in infrared to identify the vegetation (B8, B4 and B3) and in natural color that allows the approximation to the real colors of the scene (B4, B3 and B2).

2.2.4. Field Verification

Between February and November 2020, four field visits were conducted to verify the vegetation cover and determine the presence of some species in the area. During the field trips, were ensured 94 sites to corroborate some vegetation categories in which there were doubts about their classification.

2.2.5. Integration of the Inter-Ministerial Group

An inter-ministerial working group was integrated to review the progress of the technical study and to coordinate the participation of the existing communities in the study area, headed by CONANP and composed of the Secretary of Ecology and Environmental Management of the San Luis Potosi State Government (SEGAM), the Agrarian Attorney's Office of the Federal Government (PA), the National Forestry Commission (CONAFOR), and the Landscape Planning and Management Unit (UPLAMAT) of the National School of

Higher Studies Morelia Unit, Autonomous National University of Mexico (ENES-UNAM), which was in charge of preparing the study.

2.2.6. Presence of Species Verification

For flora species, from the condensed records obtained from CONABIO's database [31], their presence was corroborated during field verification visits, and a photographic record of the most physiognomic important species was obtained.

For the fauna groups, although biological collections were not carried out as such, the presence of species was recorded during the walks by direct or indirect observation. Amphibians and reptiles were identified with the help of the taxonomic keys of Flores-Villela and collaborators [46] and González-Hernandez and collaborators [47], also with the help of the book of amphibians and reptiles of the state of San Luis Potosí [48]. Bird observation was through binoculars, identification was with the support of field guides [49–51]. The presence of medium and large mammals was made by direct and indirect observation (tracks and excreta), which were identified with the support of the book by Aranda [52].

2.2.7. Environmental History

An important aspect of the methodology is the incorporation of environmental and agrarian history to understand the different uses and management of the environment in the SSM. Based on the consultation of historical archives, such as the National Agrarian Archive and the Historical Archive of San Luis Potosi State, as well as the consultation of historical cartography of the Orozco y Berra map library, it was possible to reconstruct the forms and changes in land tenure from 1900 to 2020.

## 3. Results

### 3.1. Ecosystems and Conservation Status

There are seven natural ecosystems, which occupy 82.42% of the total area (see Figure 2). In order of importance in the surface area: pine forest, xerophytic scrub, oak forest, and pine-oak forest. There are also three vegetation coverages related to human activities: agriculture, grazing areas, and artificial water bodies, in addition to areas heavily impacted by overgrazing.

Pine forest has a higher coverage with more than 32,555.21 ha. It is present throughout the zone, mainly from the center and south of the area. These forests are on high mountains, hillsides, and slopes. Most of them are composed of pure stands of *Pinus cembroides*; nevertheless, it is possible to find areas dominated by *P. hartwegii* in the highest parts of the sierra and *P. strobiformis*. With representative species of Asteraceae (*Stevia serrata*), Lamiaceae (*Salvia regla*), Cactaceae (*Opuntia robusta*), and Agavaceae (*Agave applanata*), the shrub and herbaceous strata are the most diverse.

Given the scale of the study, the scrub (with crass stems, small leaves, thorny, and rosette-shaped leaves) forms a melting pot of combinations that cannot be detected separately and therefore are considered entirely xerophytic, with little rainfall climatic condition. The xerophytes dominate a large part of the zone, being the second in importance with a little more than 34,000 ha. They are found to the north, where it borders the Chihuahua Desert, on hills and foothills, while to the south of our study area, it is mainly on foothills. *Larrea tridentata* is the dominant species, although small areas dominated by *Prosopis juliflora*; other species that dominate the physiognomy of these scrublands are *Opuntia robusta*, *Opuntia microdasys*, *O. streptacantha*, *Myrtillocactus geometrizans*, and *Echinocereus pectinatus*. Species of the Poaceae family (*Bouteloua curtipendula*) dominate the herbaceous stratum.

Oak forest occupies almost 10,000 ha. It is important to note that this includes formations dominated by oak trees in their appearance. Their distribution area extends to the south and north of the study area, in low mountains, slopes, and foothills. They are usually composed of *Quercus chihuahuensis* and *Q. resinosa*, species that physiognomically dominate the tree stratum. Species of the Asteraceae family are the best-represented in the shrub stratum.

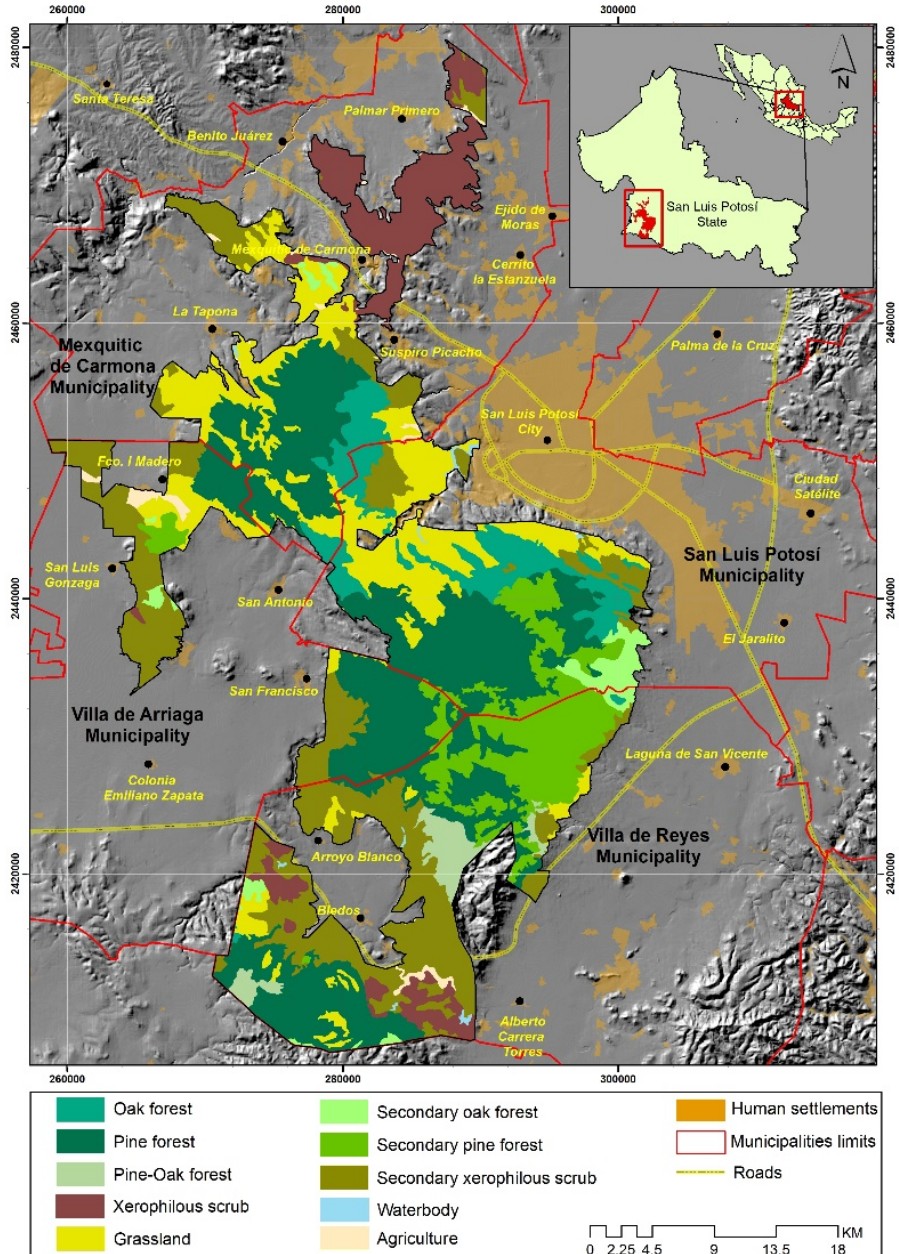

**Figure 2.** Vegetation types in the study area, in 2020.

The mixed pine-oak forest is the less extensive vegetation cover, almost 2300 ha distributed from the center to the south and east of our study area, particularly with the border of the urban area of the SLPc. They are mainly on gentle hills and foothills. The pine species mentioned above are the ones that dominate the arboreal stratum, together with *Q. obtusata*, *Q. potosina*, or even *Q. eduardi*. While the shrubs, whose composition is considerably impoverished, are dominated by species of the Asteraceae family (*Stevia lucida*).

About 67% of the area covered by the ecosystems present in the SSM is in relatively well-conserved conditions, mainly in the steeper parts of the sierra (highlands). The dynamics in the lower and middle areas are different because they are the foothills and surrounding sites with little or no slope. In addition, these areas have suffered a lot of changes for primary activities (i.e., agriculture and cattle ranching), which has caused marked alterations such as the total loss of vegetation cover, making the soils more susceptible to wind and water erosion.

### 3.2. Environmental Services

The SSM provides environmental services that benefit the surrounding population, such as carbon sequestration, mitigation of climate change effects, aquifer recharge and water infiltration, climate regulation, soil retention, flood prevention, surface runoff regulation, maintenance of biological diversity, soil formation, and conservation, pollination, maintenance of medicinal and ornamental resources, scenic beauty, recreation, and environmental research and education.

The southern portion of the PNA is located within the "Confluencia de las Huastecas" hydrological region, considered on the 110 identified priority hydrological regions (RHP) in the country [53], being areas with potential for conservation due to their high biodiversity; however, general problems were identified in all the RHP, such as overexploitation of surface and groundwater, desertification and deterioration of aquatic systems, contamination of aquifers, accelerated erosion processes caused by deforestation, modification of natural vegetation, soil loss and fires, among others [54]. In this sense, the SSM is a deep recharge area of the semi-confined aquifer of the San Luis Potosí Valley [55] that supplies the SLPc.

The creation of the SSM PNA will also promote the maintenance of the following environmental services:

Conservation of a significant area of temperate forests (pinyon pine) and xerophytic scrub, poorly represented in the country's protected natural areas.

The conservation of 55 species under some category of protection and 285 endemic species. Some relevant species are the golden eagle (*Aquila chrysaetos*), the domestic mallard duck (*Cairina moschata*), the sotol cucharilla (*Dasylirion acrotrichum*), and the Sierra laurel (*Litsea glaucescens*).

It may be an enabling area for the reintroduction of the Mexican wolf because of its characteristics in its habitat and historical distribution.

It is an important resting, feeding, and refuge area for migratory species, such as, for example, the American marten (*Lanius ludovicianus*) with a decimated population in the United States of America and Canada.

The establishment of a biological corridor between high biodiversity habitats (Gogorrón National Park, SSM, State PNA and the north of the municipality of Mexquitic) to maintain gene flow and displacement of fauna, also the conservation of the national and state cultural heritage of SLP.

### 3.3. Land Use Management

The ANP can be a milestone in the region to promote the diversification of economic activities of the populations from a sustainable perspective, such as ecotourism, the use of biodiversity through the establishment of environmental management units (UMA) for songbirds and ornamental birds (white-winged dove, quail, waterfowl, hares, rabbits, etc.). As well as the management of non-timber plants such as those belonging to the Cactaceae and Agavaceae families and the sustainable management of pinyon pine (*Pinus cembroides*) forests, which can be a non-timber resource use activity with a high economic impact on the local population.

### 3.4. Species Richness

#### 3.4.1. Flora

Within the study area, there are 99 families, 383 genera, and 735 species. The most diverse family is Asteraceae with 135 species, followed by Poaceae with 116, Cactaceae with 47, and Fabaceae with 33 species. At the generic level, Muhlenbergia had the highest number of species with 28, followed by Opuntia with 17, Salvia with 13 taxa, and Agave with 12 species. Finally, one species represents 32 families, the same for 243 genera.

3.4.2. Fauna

In general terms, the wealth of vertebrates (fish, amphibians, reptiles, birds, and mammals) represents about 33% of the total reported for SLP, in only 1.79% of the state's surface area.

Fish are the group with the lowest number of records in the SSM with only six species, close to 9% of the total recorded at the state level. Amphibians are conforming by 11 species, which remains about 3% of the national total and more than a quarter of those reported for SLP. Regarding reptiles, the 38 species reported in the SSM represent about 5% of those recorded in Mexico and 27.54% of those described at the state level.

Birds are the group with the highest number of species, recorded to 202, representing 17.99% and 37.55% of those reported for the country and the state, respectively. Mastofauna of the SSM represents about 11% of the country and 33.33% of the SLP mammals, 54 species in total. The results broken down by faunal group are given in Table 1. At this point, it should be made clear that both, for the listings and the count of species, infraspecies, varieties, subspecies and forms of a single species, were counted as a single record.

**Table 1.** Number of terrestrial vertebrate species reported for Mexico, San Luis Potosi and the Sierra de San Miguelito.

| Taxonomic Group [1] | San Luis Potosí [2] | Sierra de San Miguelito [3] |
|:---:|:---:|:---:|
| Fishes (2723) | 69 | 6 |
| Amphibians (376) | 43 | 11 |
| Reptiles (864) | 138 | 38 |
| Birds (1123) | 538 | 202 |
| Mammals (496) | 162 | 54 |

[1] Total species reported for Mexico: fish [56], amphibians [57], reptiles [58], birds [59], and mammals [60]. [2] The total number of species of each group reported for San Luis Potosi was taken from the Information compiled from the appendices and contents of the bibliography [61,62]. [3] The data presented for the Sierra de San Miguelito correspond mainly to the CONABIO database [31] and the bibliography consulted.

*3.5. Richness by Taxonomic Group*

3.5.1. Fish

The ichthyofauna reported for this group in the proposed area corresponds to six species belonging to five orders, the same number of families, and six genera that correspond to about 9% recorded for the state.

3.5.2. Amphibians

The amphibians reported in the SSM are represented by two orders, six families, eight genera, and 11 species, which correspond to 25.58% of the amphibians registered for the state of SLP.

3.5.3. Reptiles

Concerning reptiles, 38 species were reported, corresponding to 27.54% of the state total, distributed in two orders, nine families and 19 genera.

3.5.4. Birds

Avifauna is the taxonomic group with the highest specific richness in the study area, with 202 species, distributed in 18 orders, 53 families and 148 genera. The bird community constitutes a little more than 66% of the total terrestrial vertebrates reported for the SSM. In this same sense, avifauna found in the Sierra (*n* = 202) represents a significant percentage of that reported for the state (37.55%) and the country (17.99%), in a relatively small area.

### 3.5.5. Mammals

Fifty-four species of mammals have been reported in the SSM belonging to seven orders, 16 families and 39 genera. The richness of mammals in the area represents 33.33% at the state level and 10.89% in the country.

### 3.6. Endemisms

The number of endemisms by taxonomic groups is shown in Table 2. Then, the breakdown by biological groups is expressed.

**Table 2.** Endemic taxa by biological group, Sierra de San Miguelito, San Luis Potosi.

| Biological Group | Endemisms |
|------------------|-----------|
| Flora | 253 |
| Amphibians | 3 |
| Reptiles | 18 |
| Fishes | 1 |
| Birds | 2 |
| Mammals | 8 |
| Total | 285 |

### 3.6.1. Endemic Flora

The 253 endemic species of flora are distributed as follows by family. The Asteraceae family has 47 endemic species, while Cactaceae is represented by 40 species, Poaceae with 32 and Asparagaceae with 25. The remaining species belong to the Melanthiaceae family. At the generic level, Opuntia and Muhlenbergia have the highest number of species with 15 taxa each, followed by Quercus and Agave with 11 taxa.

### 3.6.2. Endemic Fauna

The 32 endemic species of fauna identified in the SSM are distributed as follows by taxonomic groups (the distribution of amphibian species was taken from [63]; while those of reptiles from [64]).

Three amphibian species were found to be endemic to the country (27.27% of those reported for the SSM) (*Ambystoma velasci*, *Dryophytes eximius*, and *Lithobates montezumae*).

Eighteen endemic reptile species were found, about half of the reptiles recorded in the study area (*Kinosternon integrum*, *Barisia ciliaris*, *Holbrookia approximans*, *Phrynosoma orbiculare*, *Sceloporus cyanogenys*, *Sceloporus dugesii*, *Sceloporus minor*, *Sceloporus parvus*, *Sceloporus scalaris*, *Sceloporus spinosus*, *Sceloporus torquatus*, *Plestiodon lynxe*, *Conopsis lineata*, *Conopsis nasus*, *Pituophis deppei*, *Rhadinaea gaigeae*, *Storeria storerioides*, *Thamnophis melanogaster*).

Among fish, only *Xenoophorus captivus* is identified as an endemic species.

Among the birds, two species are endemic (*Icterus abeillei* and *Sporophila torqueola*).

As for mammals, eight species are endemic. One gopher, one rat and six mice (*Cratogeomys goldmani*, *Dipodomys phillipsii*, *Chaetodipus lineatus*, *Chaetodipus nelsoni*, *Peromyscus difficilis*, *Peromyscus furvus*, *Peromyscus melanocarpus* and *Peromyscus melanophrys*).

### 3.7. Cultural Richness

The values of cultural richness include both the official records kept by the National Institute of Anthropology and History (INAH) and those elements that are part of the collective memory of the inhabitants of the towns adjacent to the sierra, as well as those of the visitors who travel through its trails, ravines and mountains. Part of this knowledge of the territory is made through the toponyms that exist in the area. Thus, in this section, we will include the official heritage records, the presence of sites of historical value such as the haciendas and the toponymy of the sierra.

In the Public Register of Monuments and Zones of Archaeological and Historical Monuments (RPMZAH), we have information related on the one hand to archaeological sites that are from the pre-Hispanic era and, on the other hand, to historical sites that are

after the sixteenth century. In the first group, we have a total of 10 records that correspond to the pre-Hispanic era [65]. Within these records, we have reference to the existence of cave paintings that correspond to different stages of occupation by groups of hunter-gatherers who occupied the sierra in different historical periods, and who at the time were referred to as Chichimecas, within which the most representative of the area were the Guachichil groups [66].

Similarly, the RPMZAH has a record of two historic sites, the first of which is the "Santuario del Desierto", with the category of religious architecture, as a convent dating from the eighteenth century, this site also known as the church of the desert is located west of the San Luis Potosi city, in the limits of the communities of Guadalupe Victoria and Escalerillas. This site is currently maintained as an important religious site where important festivals and pilgrimages take place. In second place, we also have the aqueduct of Cañada del lobo, which dates from the XIX century, whose importance derives in great part by supplying water to the SLPc.

In addition to the record of these sites, the presence of different haciendas can also be considered as sites of cultural relevance. In the area, there is a record of ten haciendas that were developed in the area adjacent to the SSM; these haciendas were developed from the seventeenth, eighteenth and nineteenth century, such as the following: La Tenería, Arroyos, La Pila, Jesús María, Gogorrón, Carranco, Bledos, San Francisco, El Tepetate and Jaral de Berrio. Although these haciendas are currently outside of the boundaries of the ANP proposal, they could be incorporated as part of a touristic route that highlights their historical importance in shaping the region.

In the same sense, the cultural value of toponymy can be highlighted as a sample of the processes of appropriation of the sierra by its different inhabitants. Thus, it can be mentioned that toponymy alludes to elements of the flora and fauna of the area, as well as to elements of the relief. In the review of the study area, 431 toponyms were found, which were grouped into five categories: hydrology, relief, localities, places and urban-industrial (see Table 3).

**Table 3.** Toponym category and number of records, Sierra de San Miguelito, San Luis Potosi.

| Toponym Category | Number of Records |
| --- | --- |
| Hydrology | 170 |
| Relief elements | 167 |
| Human settlements | 65 |
| Places | 24 |
| Urban-industrial | 4 |

Among the bodies of water, we can mention that reference is made to dams, rivers, streams, lagoons and water wells. Of the rivers, we can mention the following: Potosino, Españita, Carranco, San miguel, Mexquitic. As for the dams, we have the San José dam, the Cañada del Lobo, Gonzalo N. Santos, the toll, among others. On the other hand, in the streams, we have a record of 115 toponyms such as: ojo zarco, las calabacillas, las cabras, el lobo, la ordeña, las borregas, el carbonero, palma chamuscada, el difunto, los toriles, rincón de las bodas. To a large extent, we see how these names reflect the appropriation of the territory of the sierra by its inhabitants.

Regarding toponymy related to relief, some of the names refer to the names of hills, mostly the outstanding elevations, as well as mesas and ravines. In the 167 registers on this category, 92 correspond to names of hills, among which we can mention, the Potosino that stands out for being one of the highest elevations of the sierra, we can also mention the Cerro Grande that is located between the limits of the municipality of San Luis and Mexquitic. As for the allusion of the mesas, these refer to a relief formation that have among their characteristics the elevation of the hills, but have a flattened surface at the top, of these, we have 38 records, such as Mesa de los Conejos, Los Chilitos to the South, San Roque, Silva, El Jacal just to mention a few. As for the ravines, there are 7 toponyms that allude to

this form of relief such as, the Canyon of the Eagle, The Shawl, The Tiger, The Yerbabuena and The Negritas, and, in the case of the hills, there are 15 points in that category.

The cultural richness of the SSM can be observed in archeological elements such as paintings and pre-Hispanic sites, as well as historical sites such as the desert sanctuary and the presence of the region's haciendas, while also considering the relief sites that also have cultural value, such as the hills, ravines, mesas, and trails, which are clear examples of the mark left by human occupation on the landscape (Figure 3).

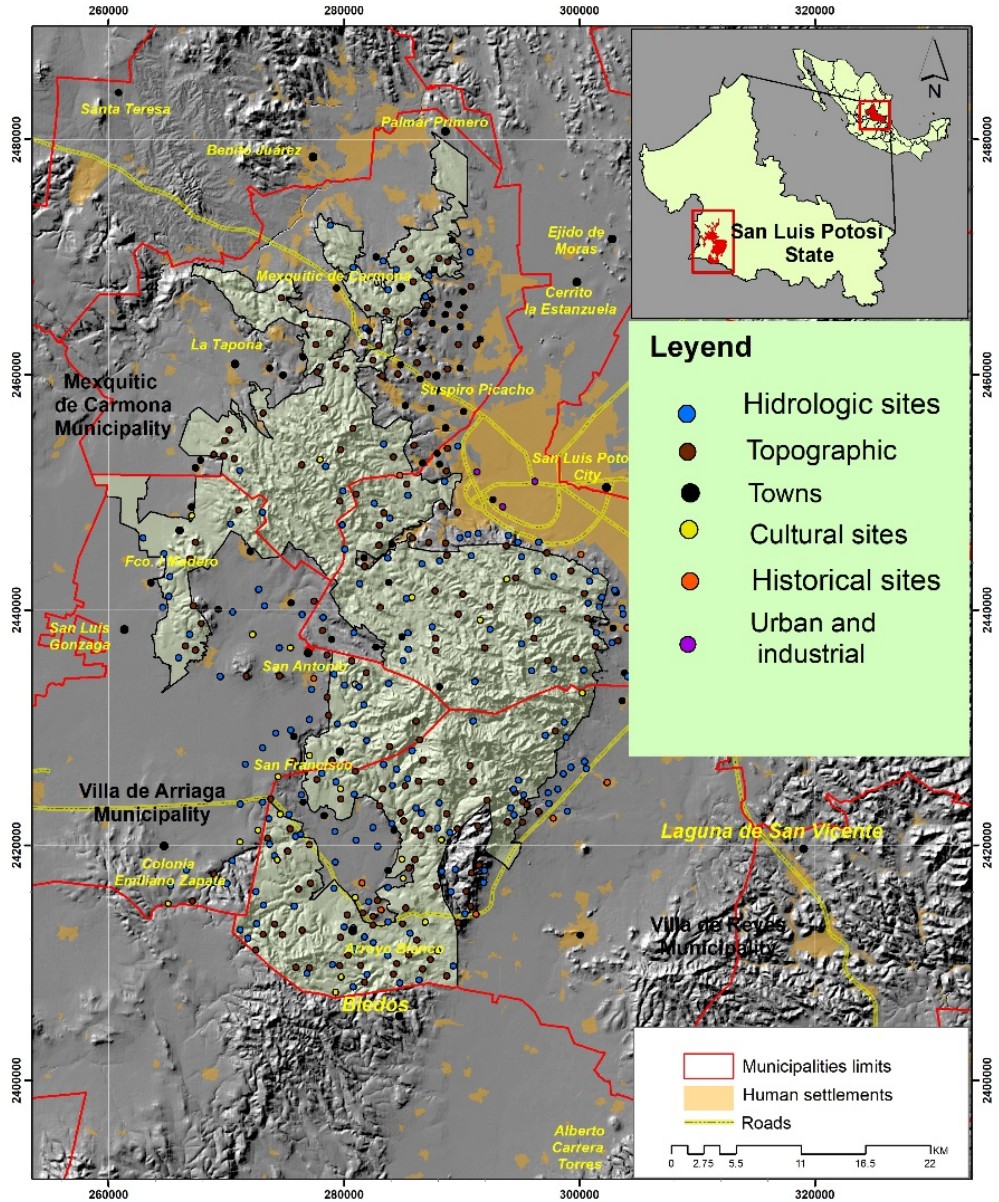

**Figure 3.** Map of cultural richness, Sierra de San Miguelito, San Luis Potosí.

*3.8. PNA Proposal*

The delimitation of the preliminary polygonal was established from the biophysical characteristics of the area presented in the previous results. The groundwork proposal of polygonal was adjusted, based on the results of a socialization process (which included participatory mapping activities) with the agrarian nuclei included, carried out between 2 and 25 October 2020, through some updates were known in changes of land tenure, the authorization of development projects by the competent authorities and the interest of some agrarian nuclei to incorporate more surface of common use.

Modification proposals were analyzed closely with the inter-ministerial technical group to identify their viability, based on the physical and biological characteristics of each particular case, current territorial management instruments that existed, conservation objects, and the disincorporation of plots from the ejido regime.

The criteria in Table 4 were applied to delimit the polygonal of the PNA, to establish the limits based on easily recognizable elements on the ground or with the information stored in the geographic information system. A vegetation and land use criterion was used from the edge of the wooded area or of xerophilous scrub recognized in the satellite image used. Physical criteria were used based on features identified on the ground, such as the course of a river or stream, a ravine, a break in the slope, or following a contour line recognized on the topographic map, economic criteria were used based on their recognition on the ground since they are communication ways (highway, bypass, road, road path) and dam boundaries. In the case of political-administrative limits, the information provided by INEGI on state and municipal limits and urban area limits was a reference, the limits of state and federal protected natural areas present in the zone were provided by the corresponding federal and state government institutions. Finally, the limits of a social nature were those supplied by the National Agrarian Registry (RAN).

**Table 4.** Criteria for the delimitation of the polygonal of the proposed Protected Natural Area, Sierra de San Miguelito, San Luis Potosi.

| Subject | Criterion |
| --- | --- |
| Environmental | Vegetation and Land use. |
| Physical | Level curves, Topography, Hydrology, Mixed, Union. |
| Economic | Communication ways, Dam. |
| Administrative/political | State PNA, Federal PNA, State limits, Municipalities limits, Urban area. |
| Social | Parcel area, Agrarian nuclei, Common use, Private property. |

The following primary zoning is proposed based on the analysis of the physical-biological characteristics of PNA and the identification of the conservation objects (Table 5): a buffer zone and three core zones.

**Table 5.** Surface of the core and buffer zones of the Protected Natural Area Sierra de San Miguelito, San Luis Potosi.

| Zone Type | Surface ha |
| --- | --- |
| Core | 24,516.4 |
| Buffer | 85,122.55 |
| Total | 109,638.95 |

The establishment of three nucleus zones is justified based on the information collected in a bibliographical manner and the field verification tours carried out from 22 to 24 July 2020, to recognize the conditions of the areas susceptible to be designated as core area and according to art. 47 Bis-1 of the General Law of Environmental Equilibrium and Environmental Protection (LGEEPA), based on the conservation of the vegetation cover, the areas that provide the principal environmental services, and the distribution of species at risk (see Figure 4).

To define the core area, the criteria were: vegetation in a good state of conservation, the presence of species at risk recorded, the core areas of state-protected natural areas, and defined areas with a conservation policy in the community land use planning. The buffer zones of the state-protected natural areas, zones affected by the 2019 fire, zones under authorized forest management, mining concessions, agricultural zones, zones with exploitation policy in the current territorial order of the municipality of San Luis Potosi, and the utilization zones established in the community land-use planning, were excluded.

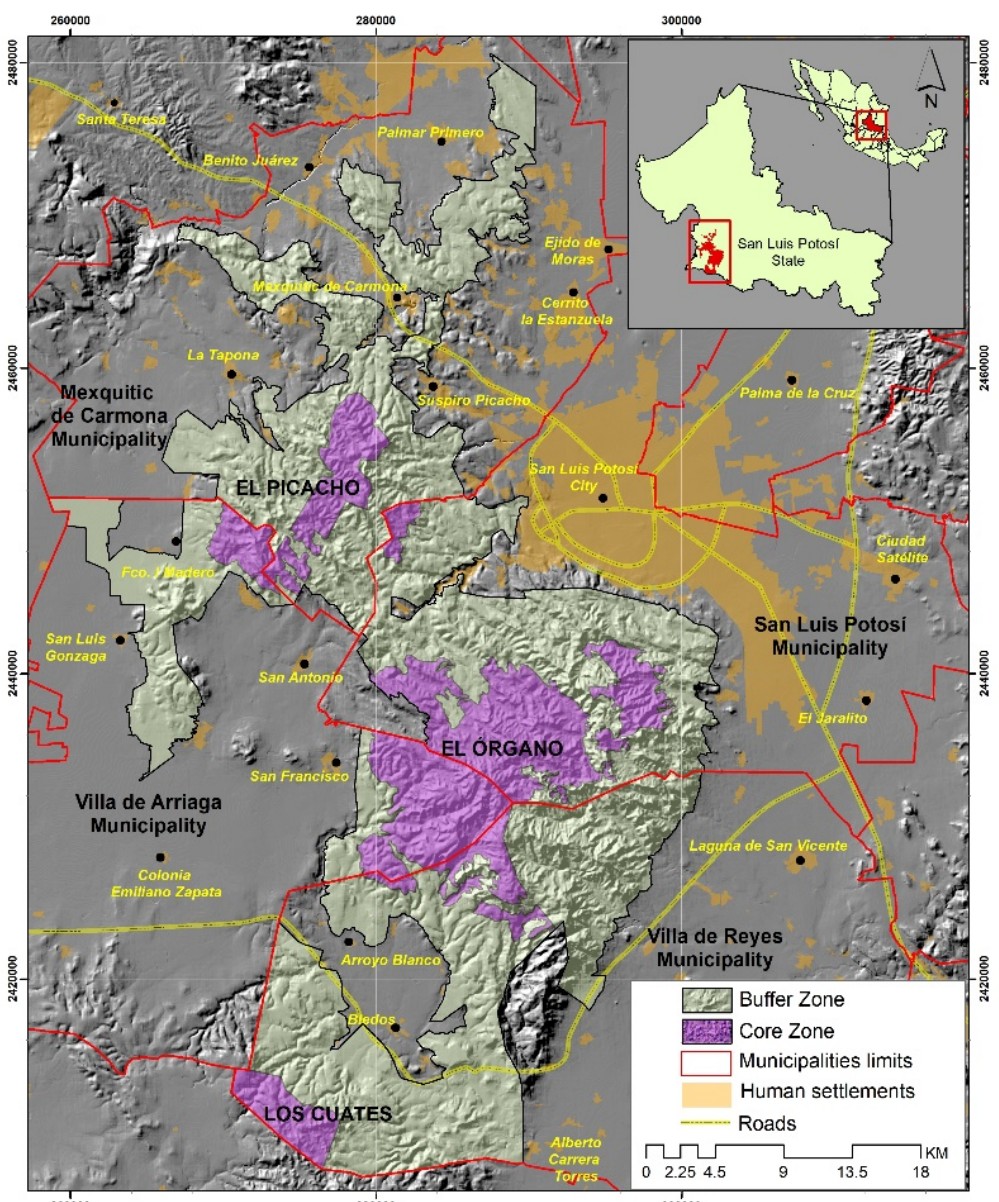

**Figure 4.** Map of core zones of the Protected Natural Area Sierra de San Miguelito, San Luis Potosi.

The vegetation and land use criterion was used based on the limit of the wooded or xerophytic scrub zone recognized in the satellite image used and verified in the field. The physical criterion used was the slope break. The economic criterion was the boundary of a mining concession. Political-administrative boundaries were used based on information provided by INEGI. The state and municipal boundaries, the federal and state governments (protected natural areas under their corresponding jurisdiction), and the social boundaries were provided by the National Agrarian Register (RAN) and the boundaries of each agrarian nucleus were used.

Below is a description of the conservation objects recognized in each of the proposed core zones.

### 3.8.1. North Core Area, El Picacho

The surface is characterized by the presence of pine forests in 87.02% of the area, with coverages ranging from 71–100%. Forest cover ranges from 41–70%, and its density is the majority to the biophysical characteristics of the area, and a lesser extent to anthropogenic impacts. The soils in this zone play the principal role because, being young, undeveloped

and stony, they favor less dense cover. The least representative categories in this zone correspond to areas with secondary xerophytic scrub, which have a cover density ranging from 11–40%; this area is northwest of polygon 2 in a transition zone between the pine and xerophytic scrub categories. Finally, there are sites with no apparent vegetation and grasslands, both categories represent only 1.2% and are placed on the peripheries of polygons 1 and 2.

In this core area, there are records of an endemic amphibian with special protection (*Lithobates montezumae*), an endemic reptile (*Conopsis nasus*), a threatened bird (*Anas platyrhynchos diazi*), two endemic mice (*Peromyscus furvus* and *Peromyscus melanocarpus*), and five species of endemic vascular plants (*Quercus chihuahuensis*, *Quercus eduardi*, *Quercus potosina*, *Salvia unicostata*, and *Schaffnerella gracilis*), which shows the good state of conservation of the habitats.

### 3.8.2. The Middle Core Area, the Organ

It is predominated by pine forest with 90.30% and densities in their covers from 71 to 100%. The biophysical characteristics of this area such as the rugged topography with steep slopes, the deep and developed soils, as well as climatic conditions, propitiate conditions for the development of temperate pine-oak forests. On the other hand, a limited human presence has favored the conservation of representative ecosystems, such as stone pine forests or pine-oak forests. These mixed forests represent 8.1% and have coverage of 41–70% and, as in the core area of El Picacho, these forests are in areas with shallow and with high stoniness soils, which contribute to vegetation with lower densities. Furthermore, some of these forests (especially where pine trees predominate) have suffered from fires, so some coverage has been affected. Finally, 1.62% of the surface corresponds to secondary xerophilic scrub with densities of 11–40%.

In this core zone, there are records of an amphibian in special protection (*Lithobates berlandieri*), an endemic tortoise and in special protection (*Kinosternon integrum*), four endemic lizards (*Barisia ciliaris*, *Sceloporus minor*, *Sceloporus scalaris*, and *Sceloporus spinosus*), a lizard in special protection (*Sceloporus grammicus*), an endemic and endangered snake (*Pituophis deppei*), a threatened snake (*Thamnophis eques*), a sparrowhawk in special protection (*Accipiter striatus*), a semi-endemic calandria (*Icterus parisorum*), a quasi- endemic bulrush (*Junco phaeonus*), a pinyon pine in special protection (*Pinus cembroides bicolor*), and four endemic vascular plants (*Quercus crassifolia*, *Seymeria virgata*, *Sotoa confusa*, and *Stenocactus phyllacanthus*). All this richness and diversity is a true reflection of the good state of conservation found in this core area.

### 3.8.3. South Core Area, Los Cuates

It is located on the limits of the San Luis Potosi state and Guanajuato state, being part of an island of vegetation shared by both states. Pine forests (71.9%) and mixed pine-oak forests (22.9%) are identified, both with coverage of 71–100%. In 4.2% exist presence of xerophilous scrub of the core area Los Cuates and is characterized by low density (predominance of cacti) and, to a lesser extent (0.9%), is identified as a category of secondary oak forest with low coverage.

There are fewer species records in this core area, but this is due to the limited sampling effort. However, there is a record of the Red-dotted Toad (*Anaxyrus punctatus*).

In general terms, the vegetation of the three core zones is mostly temperate forests with coverage of 70–100%, which in terms of vegetation expresses an important degree of conservation that positively impacts the different cycles and natural processes, since they can be classified as habitats in a good state of conservation. These forests allow the stability of the soils, favor water recharge zones, are a refuge for wild fauna, contribute to the regulation of the climate, makeup landscapes with the high scenic value among other environmental services that benefit not only the inhabitants of the closer populations, but are also a benefit on a regional, national and global level. For these reasons, their conservation is essential, especially in a region where these ecosystems are threatened by

various activities, such as livestock, agriculture, or the expansion of human settlements, and their designation as core areas is fully justified.

## 4. Discussion

The term biodiversity conservation refers to human actions that seek to protect samples of nature—biotypes, species, ecosystems, landscapes—from human actions, and also refers to the sustainable use of natural resources [67]. To protect natural and cultural heritage, many countries have adopted the model of natural protected areas proposed by various international conservation conventions [68].

San Luis Potosi has 14 state and 5 federal PNA [69]. The flora and fauna protection area "Sierra de San Miguelito" is a proposal of federal character that contemplates within its limits the natural area called "Reserva Estatal Sierra de San Miguelito" that was decreed in 2018 and has an area of 12,613.47 ha, it also borders with two more protected areas. One of them is the state urban park "Paseo de la Presa" decreed in 1996 with an area of 344.02 ha [70] and the national park "Gogorrón" decreed in 1936 which has an area of 38,010 ha [71]. In environmental terms, this is beneficial because the incorporation of this new natural area will allow the connectivity of an extensive biological corridor between habitats with high biodiversity. This type of corridor has been implemented in different parts of the world and has been very successful because, by improving connectivity, species, landscapes and ecosystem services are protected [72–75].

In this sense, it has been demonstrated that the conservation of species, ecosystems and habitats within protected areas can be more efficient if they are functionally connected or integrated into a broader landscape [76–78]. Therefore, when establishing a natural protected area, its functioning, size, connectivity and representativeness must be considered and, in this way, avoid problems associated with isolated areas without possibilities of species recolonization or areas that are too small and do not have enough space to maintain populations with ample space requirements (e.g., predators such as eagles) [79].

The preservation and protection of the flora and fauna richness is a priority, since the sierra area has a high conservation potential because 18.3% of the flora and 32.7% of the terrestrial vertebrates recorded for the SLP are found in the SSM, in an area that represents only 1.79% of the state surface. In addition, just over 5% of this diversity is under some category of protection in the NOM-059-SEMARNAT-2010 [80,81] and more than a quarter are endemic (*n* = 285).

The main threats posed by the SSM are the fragmentation and loss of ecosystems, the contamination of rivers and bodies of water, hunting, and the illegal trade of species; mainly due to the change in land use for human settlements, the growth of the agricultural frontier and extensive cattle ranching.

The establishment of the SSM as a protected natural area acquires greater relevance, not only for its contribution to biodiversity conservation but also for the environmental services. It provides to human well-being and productive activities in the region, in such a way that its conservation can be the guiding principle for the sustainable development of human populations that inhabit the zone surrounding the area.

The temperate pine (*Pinus cembroides*) and oak forests (*Quercus* spp.), together with the different xerophytic scrub associations present in the proposed PNA, support a great diversity of species, high numbers of species under some category of protection and endemic species, concerning the area covered. Together with the low density of the human population, these are factors that make this area ideal for the establishment of a protected natural area, in addition to all the ecosystem services that it provides.

The habitats present in the area (pine forest, oak forest, mixed forest, and different associations of xerophilous scrub) are of great importance for the conservation of the existing biological diversity, its spatial and temporal continuity, the development of its populations, as well as the ecological and biological processes that occur in these ecosystems [54,82].

Among the main environmental services identified provided by the area proposed to be declared such as ANP, we have:



- The protection of 49,055.41 ha of soils with high and medium erodibility [83,84];
- Soil retention in 85,290.7 ha of steep slopes;
- The protection of three hydrographic basins (San José-Los Pilares, Tamuín River and SLP Dam) [16,17,85];
- Maintain the recharge of three aquifers (San Luís Potosi 78.1 $Mm^3$/year, Jaral de Berrios-Villa de Reyes 132.1 $Mm^3$/year and Villa de Arriaga 4.8 $Mm^3$/year) [21,86–89];
- Provide drinking water for domestic, agricultural, livestock, and industrial use (more than 1,000,000 domestic users in the four municipalities, 10,383 ha of irrigated agriculture, more than 8000 head of cattle, and the entire industrial zone of SLP) [17,21,87,88];
- Maintain the water supply in 33 dams provided by runoff from the proposed PNA (18 within the proposed polygon) [15,17,86];
- Maintain timber forest uses (52,269 ha with forest management, production of 2579 $m^3$ per year) and non-timber forest 86 authorized uses for plants, fibers, and forest land) [39];
- Fixation in plant biomass of at least 500 thousand tons of carbon per year [36];
- The sustainable use of the area through ecotourism and non-timber forest resources, ecological education and recreational activities could be the cornerstones of sustainable rural development for the agrarian nuclei involved in the proposed polygonal area.

On the other hand, in addition to its great natural richness, Mexico is characterized by one of the greatest cultural diversities in the world. The proposed flora and fauna protection area contributes in a relevant way to the objectives of the "National Program of Protected Natural Areas 2020–2024", which highlights the importance of protecting sites such as archeological zones of pre-Hispanic cultures, as well as cave paintings, historic buildings and relevant architectural ruins [90].

Among the successes of the work carried out, the establishment of the intergovernmental working group provided legal and institutional certainty to the process, as well as consensus in the validation of the technical results obtained between the participating institutions.

The results of the socialization generated from a participatory planning process (participatory mapping) made it possible to modify the initial proposal in conjunction with the interested agrarian nuclei, generating greater acceptance of the proposal. These types of methodologies have already been recognized as very successful in biodiversity conservation [91–93].

The methodological basis used in this work has shown satisfactory results in other conservation processes based on the landscape planning approach [26,94–96], or to promote sustainable landscape management [28].

PNA, together with other protection schemes, are fundamental for the conservation of the landscape, which within its multiple meanings has also been considered an environmental resource [97–99], which has determined its inclusion in environmental management and protection instruments, due to its territorial component [98]. Therefore, landscape planning through different instruments of public policy and social action are indispensable for the integration of conservation, restoration, and sustainable use of natural resources [100].

The result of this research was the basis for initiating the negotiation process with the agrarian nuclei, developed during the year 2021 by the intergovernmental working group headed by CONANP, for the formal creation of the PNA.

The main limitations of the work were the restrictions on field work and participatory planning meetings due to the health contingency caused by the SARS-CoV2 pandemic.

## 5. Conclusions

The SSM area has a high diversity of species, a little more than 18% of the flora and nearly 33% of the fauna reported at the state level, which means an area of less than 2% of the state's total. In addition, 5% (*n* = 55) of the biological richness reported in the SSM is under some category of protection in the NOM-059-SEMARNAT-2010, and 27% is endemic to the country (*n* = 285).

In this zone, the presence of xerophytic scrub and temperate forest ecosystems is remarkable, that provide environmental services to both the SLPc and the surrounding

population centers; however, these services are threatened by various human activities, such as the expansion of human settlements, the agricultural frontier, grazing, and forest fires.

In hydrological terms, the area is very important because it recharges the aquifers that supply the SLPc and the infiltration process prevents flooding; due to the presence of a large number of natural and artificial bodies of water, it is an important resting, feeding and refuge area for migratory species. It also allows carbon sequestration, helps mitigate the effects of climate change through climate regulation and contributes to soil retention.

The establishment of the PNA could be very beneficial in the region because it will allow the diversification of economic activities of the population to be promoted from a sustainable perspective. For example, ecotourism, the use of biodiversity through the establishment of environmental management units (UMA), as well as the management of non-timber plants such as those belonging to the Cactaceae and Agavaceae families, or the sustainable management of pinyon pine forests (*Pinus cembroides*), which can be a non-timber resource use activity with a high economic impact on the local population. On the other hand, its establishment can favor research, environmental education and recreation, as well as the preservation of cultural and landscape heritage, which are the basis for territorial sustainability.

**Author Contributions:** Conceptualization, G.A.H., F.A.R. and L.F.A.; formal analysis, F.A.R., L.S., L.F.A., J.M. and J.F.S.; investigation, G.A.H., F.A.R., L.F.A. and J.F.S.; methodology, G.A.H., L.S., J.M. and J.F.S.; project administration, F.A.R.; software, L.S.; supervision, F.A.R.; validation, G.A.H., L.S., L.F.A. and J.M.; visualization, L.S.; writing—original draft, G.A.H., F.A.R., L.S., L.F.A., J.M. and J.F.S.; writing—review and editing, F.A.R. All authors have read and agreed to the published version of the manuscript.

**Funding:** This research was funded by CONANP to San Luis Potosi State Government in 2019.

**Institutional Review Board Statement:** Not applicable.

**Informed Consent Statement:** Not applicable.

**Data Availability Statement:** All information used and generated during the technical study is of a public nature and is available for consultation at CONANP headquarters, Mexico City, Mexico.

**Acknowledgments:** We acknowledge the support of the inter-ministerial group in the preparation of the technical study, and, in particular, the Secretariat of Ecology and Environmental Management (SEGAM) of the San Luis Potosí state government. We thank Andrea Belen Cárdenas Pantoja for her support in the translation of the manuscript.

**Conflicts of Interest:** The authors declare no conflict of interest. The funders had no role in the design of the study; in the collection, analyses, or interpretation of data; in the writing of the manuscript, or in the decision to publish the results.

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
