# Peer review of "Landscape Planning for Conservation: The Case of the Flora and Fauna Protection Area “Sierra de San Miguelito”, San Luis Potosi, Mexico"

_diversity, doi:10.3390/d14010025_

Round 1
Reviewer 1 Report
The article “Landscape planning for conservation. The case of the flora and fauna protection area "Sierra de San Miguelito", San Luis Potosi, Mexico” is very interesting because it brings together a set of information that support the proposal for the creation of a protected area. The document is well written and is suitable for publishing in the Diversity journal, however it needs some improvement before proceeding with the publication process.
Line 4 - Titles do not need a full stop.
In the Introduction, the brief information on the characterization of the vegetation cover should be transferred to the topic Area of study.
The Materials and Methods topic is well structured and presents the ideas clearly. However, I suggest that the authors add some information in the subtopic “Study area”, such as climate data, where they refer to the minimum, maximum and measured temperatures, as well as the average annual precipitation of one or more points. Authors may also opt for a bioclimatic characterization: DOI: 10.5616/gg110001. Since this is a proposal for a protected area, where there is direct and indirect human intervention, the main activities carried out in the study area must be presented and, if possible, some quantitative data. This will help the reader to know and better understand the purpose of the work.
Line 95 - I believe the displayed legend is incomplete, as the map has other colors and textures that are not classified.
Line 97 - the subtopic “2.2. Materials and Methods” should not be repeated with the topic “2. Materials and methods". I suggest changing it to “2.2. Data collection".
The results data do not raise doubts, however, the title of the point “3.4.1. Flora” is repeated with point “3.6.1. Flora". This must not happen.
The dimension of the captions in figures 3 and 4 should be increased, in order to improve their reading.
In the discussion, the authors are supposed to compare the results of their study with other similar works. What other proposals for protected areas have followed this path and what are the benefits of protecting this area should be presented here. However, what is presented is a summary of the main characteristics of the studied area and without reference to other studies. Both in the Introduction topic and in the Discussion, I miss the consultation of articles published more recently.
Taking into account that the article deals with landscape planning, I suggest that the authors enrich the bibliographical references. Current information is important to fuel the authors' ideas, such as:
https://doi.org/10.1016/j.jnc.2018.08.005
https://doi.org/10.3390/fire3040065
https://doi.org/10.1007/s11355-021-00467-6
Author Response
All the comments and proposals made were addressed.

Reviewer 2 Report
Dear Authors,
I find your proposal very interesting being related to the analysis of the existence priority elements of biological and ecological importance for conservation in a region of Mexic in order to justify the creation of a protected natural area under federal jurisdiction. I proposed several recommendations:
-Abstract: to state more clearly the aim of the study
-Introduction: I suggest you to include in the Introduction section information related to the implications of the study and structure of the paper. The theoretical background, could be extended with some ideas.
The description of case study is too detailed in the introduction. A part of the information can be introduced to the section 2.1 Study area.
Methodology: Can you mention several aspects related to the novelty elements of the methodology, if any?
-Discussion section: you should highlight the significant contribution of the study. Several results obtained in the study should be analyzed by comparison with other similar studies. -you should mention the limitations of the study and future directions of research.
Author Response
All the comments and proposals made were addressed

Round 2
Reviewer 1 Report
Although the authors did not respond to the questions posed point by point, the changes presented in the document demonstrate that they accepted the comments and modified the text accordingly.
As all the improvements are in the text, I consider that the article "Landscape planning for conservation - The case of flora and fauna protection area "Sierra de San Miguelito" in San Luis Potosi (Mexico)" can continue with the process of publication in the journal Diversity.
I just leave as a last comment that titles usually don't need a full stop, but I leave this to the editors and authors for consideration.